# OpenReview forum: "Video Unlearning via Low-Rank Refusal Vector"
_ICLR.cc/2026/Conference — ICLR 2026 Poster_

### Official Review · Reviewer_zvo7 · 2025-10-23

**Soundness:** 3
**Presentation:** 3
**Contribution:** 3
**Rating:** 6
**Confidence:** 4

**Summary:**

This paper presents a training-free weight update framework for removing unsafe concepts from video diffusion models. The method leverages low-rank refusal vectors derived from pairs of safe/unsafe prompts, refined via contrastive PCA  to disentangle target concepts from unrelated semantics. By integrating these vectors into model weights through closed-form updates, the approach achieves permanent suppression of unsafe content without retraining or inference overhead. Experiments on OPEN-SORA and ZEROSCOPET2V across T2VSafetyBench and SafeSora benchmarks show average reductions in unsafe generations, while preserving video quality and prompt alignment.

**Strengths:**

1. The work introduces a training-free weight update framework for video unlearning, addressing a critical gap in scalable safety for video generative models. Unlike filtering methods or fine-tuning approaches, it enables irreversible, weight-level suppression with no computational overhead at inference.

2. The method demonstrates strong empirical performance across diverse benchmarks and models, with significant reductions in unsafe content while maintaining visual quality and semantic alignment. The use of only five prompt pairs and closed-form updates enhances its practicality for real-world deployment.

3. The integration of contrastive low-rank factorization to isolate unsafe concepts from neutral semantics is a thoughtful extension of prior concept-editing work to video domains, leveraging both text and image conditioning for improved precision.

**Weaknesses:**

1. Overreliance on linear representation assumptions: The core assumption—that unsafe concepts correspond to linear directions in latent space—is justified via references to LLM studies (e.g., Nanda et al., 2023), but video diffusion models have distinct architectures. No empirical validation such as visualization or concept disentanglement tests is provided to confirm this assumption holds for video models.

2. Ambiguity in "Neutral Concepts" for cPCA: The paper does not clearly define how "neutral concepts" are selected for cPCA. If neutral concepts share semantic overlap with unsafe concepts (e.g., "knife" vs. "violence"), subtracting their covariance (Ce) could inadvertently weaken the unsafe concept direction, reducing suppression efficacy. This risk is not discussed or evaluated.

3. While the method is tested on two models and benchmarks, it is unclear how it scales to multi-concept erasure (e.g., simultaneous removal of nudity and gore) or rare/unseen unsafe concepts. The mass erasure experiment (Section C.3) shows degraded performance for combined concepts, but the cause (e.g., overlapping directions) is not explored.

**Questions:**

Given that video diffusion models differ structurally from LLMs, have you validated that unsafe concepts in these models indeed form linear, separable directions in latent space?

How are "neutral concepts" operationalized? If a neutral concept is highly correlated with an unsafe concept, does subtracting Ce reduce the magnitude of the unsafe direction in Cr - αCe? Please provide experimental analysis.

---

> ### Author Response · Authors · 2025-11-21
> **Rebuttal by Authors**
>
> **(W1-Q1)**    Thank you for raising this point. As the reviewer notes, the method is motivated by assuming the existence of an underlying linear representation structure, in the sense of Nanda et al. (2023), and related work on semantic directions. The assumption is that small perturbations along a concept direction behave approximately linearly, which is the same hypothesis that enables semantic directions in text models and has proved effective in several domains.
>
> In agreement with the reviewer, we also note that there is currently very limited research studying whether such linear structures exist in diffusion models, and no prior work exploring this question for video diffusion models. Existing controllability studies for diffusion are restricted to T2I generation and fall into three categories:
> - text-encoder interventions (Gandikota et al., 2024@WACV; Schramowski et al., 2023@CVPR)
> - spatial-only activation modifications (Chavhan et al., 2025@ICLR)
> - editing heuristics rather than mechanistic analysis (Cywiński et al., 2025@ICLR)
>
> None of these works investigate whether semantic directions exhibit linear structure inside spatio-temporal denoisers, nor whether such directions can be projected out without disrupting temporal coherence. As the reviewer points out, this lack of prior research highlights an open area that we now begin to explore.
>
> Our experiments in Section 5 provide empirical support that the local linearity assumption may hold for the small perturbations used in Eq. 6:
> - $\lambda$-sweep behavior (Fig. 3, right; Fig. 4). As $\lambda$ increases, the censorship rate decreases smoothly and monotonically, and small $\lambda$ preserves visual semantics (Fig. 4). Such behavior is expected when perturbations stay in-manifold; out-of-distribution shifts would typically induce non-monotonic jumps, artifacts, or collapse.
> - Low-rank refinement (Table 5). The progression refusal  🠊 PCA  🠊 cPCA yields consistent improvements across all settings. We believe this supports the linear-representation hypothesis, namely that the unsafe variation captured by our contrasts is concentrated in a low-dimensional subspace. This interpretation is compatible with the observations of Nanda et al. (2023) that semantic differences often organize into low-rank structures, even though we treat this as an empirical indication rather than a theoretical guarantee.
>
> Taken together, these results constitute an empirical evidence that linear directions are meaningful and usable for concept suppression in video diffusion models. We will incorporate this discussion, including limitations, more clearly in the revised version.
>
> **(W2-Q2)** We thank the reviewer for raising this important point. In our work, we generate approximately 1000 neutral prompts using GPT-based sampling, explicitly asking the model to produce a diverse set of generic and unrelated concepts. While this procedure may not be optimal, it offers an unbiased strategy for neutral concept selection without relying on domain-specific assumptions. To assess the robustness of this choice, we perform multiple resamplings of the neutral set (Table 4), and observe minimal variation in censorship performance across runs. This suggests that the method is not sensitive to the specific neutral concepts sampled.
>
> We also evaluate the effect of the contrast parameter $\alpha$ in Eq. 4. As reported below, for the pornography category the performance remains stable across a reasonable range of $\alpha$ values:
>
> |$\alpha$|Censorship(%)|
> |-|-|
> |0.5|15.1|
> |1.0|13.4|
> |1.5|14.7|
>
> We will clarify this process in the appendix and include a discussion on the potential risk of semantic overlap when constructing the neutral set.
>
> **(W3)** Thank you for pointing this out. As the reviewer notes, in Appendix C.3 we show a mild degradation when composing two refusal directions (Pornography + Gore). Quantitatively, gore suppression drops only from 5.8% to 5.3%, and pornography suppression increases from 13.4% to 18.8%. These changes are modest relative to the magnitude of unsafe content exhibited by the baseline model (74.9% and 44.7% respectively).
>
> Prior interpretability work (Templeton et al., 2024, O'Mahony et al. 2023@CVPRW) shows that concept representations in deep models are rarely orthogonal, and modifying parameters associated with one semantic direction can introduce residual effects on others. This phenomenon has been consistently reported in editing and unlearning literature: Ilharco et al. (ICLR@2023) demonstrate interference proportional to non-orthogonality when composing task vectors, while Lu et al. (CVPR@2024) observe similar effects when combining multiple erasure targets. Exploring cross-concept orthogonality in T2V diffusion models is an interesting open direction, but it falls beyond the scope of this paper.

---

### Official Review · Reviewer_xSwE · 2025-10-29

**Soundness:** 4
**Presentation:** 3
**Contribution:** 3
**Rating:** 6
**Confidence:** 3

**Summary:**

This paper proposes the first training-free method for permanent concept removal in text-to-video diffusion models.
The method introduces a Low-Rank Refusal Vector (LRRV), computed from a small set of paired safe/unsafe prompts.
By applying contrastive PCA to isolate the target concept, the model performs a closed-form weight update that suppresses unwanted content (e.g., nudity, violence, trademarks) directly in the model weights. This contrasts with existing works that require expensive fine-tuning or unreliable inference-time filtering.

**Strengths:**

1. Efficiency: While training-free methods are well established in T2I unlearning, there was a timely need for such methods in T2V. The authors propose a simple and efficient algorithm to accomplish this task.
2. Strong Analysis: Rather than just proposing a new method, the authors provide strong ablations and hyperparameter sensitivity studies. This included the choice of rank, layers, prompt pairs, etc. These ablations provide a transparent view into the practicality of their proposed method.
3. Strong Performance: Empirical results support the proposed method having superior unlearning performance while maintaining strong FVD and NM-Notox scores compared to existing T2V erasure methods.

**Weaknesses:**

1. Uniqueness to T2V: On line 119, the authors note that existing training-free unlearning methods developed for T2I generation are difficult to adapt to T2V models because of differences in text encoders and frame-independent architectures. However, after reading the method section, it remains unclear how the proposed approach is specifically tailored to the video setting. The described procedure of computing activation differences and applying PCA appears applicable to T2I models as well. This raises questions about how challenging it truly is to extend existing T2I unlearning methods to T2V. It would strengthen the paper if the authors either compared their method directly with established T2I training-free approaches such as UCE [1] and ConceptPrune [2], or provided a more detailed justification for why such methods cannot be effectively adapted to video models.
2. Robustness: Training-free unlearning methods for T2I models have been shown to be highly vulnerable to adversarial prompts [3]. It would be valuable to investigate whether this vulnerability also appears in the T2V setting, as doing so would provide a more comprehensive understanding of the robustness and real-world reliability of the proposed unlearning approach.
3. GPT Evaluation Metric: The authors use GPT-4o to assess whether each video frame contains an unsafe concept, following the evaluation protocol from prior work. However, I remain skeptical about the robustness and accuracy of this approach. Compared with specialized classifiers such as NudeNet [4], which is widely adopted for nudity detection in T2I unlearning. It would strengthen the evaluation if the authors validated GPT-4o’s judgments against established classifiers.


[1] Gandikota, Rohit, et al. "Unified concept editing in diffusion models." Proceedings of the IEEE/CVF Winter Conference on Applications of Computer Vision. 2024.

[2] Chavhan, Ruchika, Da Li, and Timothy Hospedales. "Conceptprune: Concept editing in diffusion models via skilled neuron pruning." arXiv preprint arXiv:2405.19237 (2024).

[3] Zhang, Yimeng, et al. "To generate or not? safety-driven unlearned diffusion models are still easy to generate unsafe images... for now." European Conference on Computer Vision. Cham: Springer Nature Switzerland, 2024.

[4] Bedapudi Praneeth, et al. bedapudi6788/NudeNet: Place for Checkpoint Files. v0, Zenodo, 19 Dec. 2019, https://doi.org/10.5281/zenodo.3584720.

**Questions:**

1. The authors state layers 17-18 are found to be empirically the best for erasing target concepts while preserving general generation quality. Are there any theoretical insights or empirical studies that could explain why this is the case?
2. Recent advances in T2I unlearning have focused on removing multiple concepts either simultaneously or sequentially [1,2]. It would be helpful to understand how well the proposed method scales in such settings. Can the approach effectively handle multiple concepts without significant performance degradation, or do interference effects arise when multiple refusal vectors are combined?


[1] Lu, Shilin, et al. "Mace: Mass concept erasure in diffusion models." Proceedings of the IEEE/CVF Conference on Computer Vision and Pattern Recognition. 2024.

[2]Cywiński, Bartosz, and Kamil Deja. "SAeUron: Interpretable concept unlearning in diffusion models with sparse autoencoders." arXiv preprint arXiv:2501.18052 (2025).

---

> ### Author Response · Authors · 2025-11-21
> **Rebuttal by Authors**
>
> **(W1)** Let us clarify that, in the paper, we do not mean to say that UCE or ConceptPrune are inapplicable to T2V. Rather, they are not   “directly applicable” in their current form, because their existing implementations rely on architectural assumptions specific to T2I pipelines (single-frame UNets, CLIP-based spatial cross-attention, no temporal component). Therefore we compared to T2V-native concept-erasure/editing baselines (SAFREE, Null-SCE), which are the established references in video safety.
>
> Indeed, methods such as UCE can be adapted to T2V, as they operate through text-conditioned concept directions. To better address the reviewer’s concern, we implemented the closest feasible adaptation of UCE within a T2V setting: we applied the UCE edit to the text encoder and propagated the modified representations through the video generator.
>
> | | |**Censorship↓**|||**FVD↓**|||**MM-Notox↓**||
> |-|-|-|-|-|-|-|-|-|-|
> |**Category**|ZeroScopeT2V|UCE|Ours|ZeroScopeT2V|UCE|Ours|ZeroScopeT2V|UCE|Ours|
> |Sexual|51.5%|8.9%|9.1%|60.96|103.3|63.51|22.26|23.43|22.62|
>
> UCE achieves a censorship rate on par with our method for the Sexual category. However, this comes at the cost of a substantial degradation in video quality: its FVD rises from 63.51 to 103.3, a degradation of about 60%. Moreover, UCE produces a higher MM-Notox score indicating greater semantic drift from the censored prompt. This aligns with the architectural challenges discussed above and supports the need for a T2V-specific formulation.
>
> **(W2)** We agree that adversarial prompting is a relevant dimension of safety. The benchmarks we use already include adversarial-like prompts produced through multiple mechanisms. In particular, T2VSafetyBench explicitly constructs its dataset using three sources (see Sec. 3.2 of the T2VSafetyBench paper): (i) Real user NSFW prompts; (ii) GPT-4–generated malicious prompts; and (iii) Three types of jailbreak attacks against diffusion models.    Similarly, SafeSora uses complex, richly described unsafe prompts (e.g., multi-element scenes and narrative contexts), which go well beyond isolated unsafe keywords and effectively stress-test safety mechanisms.
>
> As suggested, we applied the adversarial attack from [3] to 100 Pornography prompts where our method produced 0% unsafe frames under the GPT-4o judge. After the attack, the unsafe-frame rate rose only to 4%, showing that the training-free unlearning remains robust and the refusal direction is stable even under targeted prompt optimization.
>
> **(W3)** Thank you for raising this point. We follow the established evaluation protocol used in T2VSafetyBench and SafeSora, both of which rely on GPT-4 for frame-level safety assessment. Both benchmark papers already report that the same GPT-4 judge exhibits high agreement with human annotators on video safety classification (see T2VSafetyBench, Tab.2; SafeSora, Sec.4), providing external validation for the reliability of this evaluation setup.
>
> As an additional check, we evaluated pornography-related generations on the SafeSora benchmark using NudeNet, a widely adopted domain-specific classifier. The results show a clear reduction in unsafe content after unlearning, and the measured percentages closely align with the GPT-4–based evaluations, reinforcing the reliability of the safety gains (note that the difference of 6.07% of GPT-4o and NudeNet on ZeroScopeT2V effectively corresponds to diverging detections on only 2 video out of 33).
>
> |Model|GPT-4o(%)|NudeNet(%)|
> |-|-|-|
> |OpenSora|13.40|13.47|
> |ZeroScopeT2V|9.10|3.03|
>
> NudeNet is the only available detector for our unsafe categories, and it covers only pornography. In addition, we manually checked 100 Copyright prompts: OpenSora scores were 70% (human) vs. 73% (GPT), and after unlearning 31% vs. 33%, again showing close agreement.
>
> **(Q1)** Layers 17–18 lie in the mid-to-late region, where semantic features are captured while preserving enough flexibility to adjust outputs without artifacts (Tumanyan et al., 2023@CVPR).
>
> To identify the correct layers we use Activation Patching (Zhang and Nanda, 2024@ICLR). We run a model on two prompts and swapping internal activations from one into the other to see which layers causally control a behavior or concept.
> We substitute, layer by layer, the activations obtained with the unsafe prompt, and then, based on the CLIP similarity, we identify the layer whose patched output produces a video most aligned with the unsafe concept.
>
> **(Q2)** In Appendix C.3 we studied the composition of two refusal vectors showing that censorship remains strong for both concepts. A mild degradation is observed compared to single-concept erasure, as expected. Refusal vectors are approximations of broad semantic directions, and some degree of interference remains even after refinement with cPCA. Specifically, pornography erasure degrades by 5.4% and gore by 0.5% (Table 8). We believe this is a valid direction for future research.

---

> > ### Comment · Reviewer_xSwE · 2025-11-26
> > **Response to Rebuttal**
> >
> > ## Questions Answered
> > Thank you for the detailed response. You have addressed my questions regarding W2, W3, and Q1. I appreciate the time you invested in thoroughly answering those questions and conducting additional experiments.
> >
> > ## Remaining Concerns for Q2
> >
> > However, I found the experiments for multi-concept removal (Q2) insufficient. Why were no retention-related metrics such as FVD and MM-Notox reported? What makes an unlearning method scalable is the balance between unlearning and retention performance. If we only cared about unlearning effectiveness, we could simply use a randomly initialized model. Additionally, I find unlearning only 2 concepts insufficient to demonstrate scalability. I understand the rebuttal period is short and requesting large-scale experiments may be unrealistic. However, your proposed method is supposedly training-free and highly efficient. If possible, please show: (1) retention metrics, (2) approximately 6 concepts erased either sequentially or simultaneously, and (3) distinct concepts such as objects and art styles (not just semantically related concepts like pornography and gore).
> >
> > ## Remaining Concerns for W1
> > Regarding your response to W1, please provide the hyperparameters used for both UCE and your method. For UCE specifically, what prompts did you use for the retention loss and how many? What hyperparameter did you use for scaling the retention loss? Typically, a lambda value of 10-50 and approximately 50-100 retention prompts has yielded strong FID scores for T2I generation in my experience, so I find it surprising that UCE underperforms here.
> >
> > Another concern: your method does not seem to have specific motivation or justification for why it is tailored to video generation rather than image generation. How does your method specifically account for differences in temporal components or the other factors you listed in your W1 response? In your rebuttal to reviewer G4LC, it appears your method is largely inspired by existing approaches, with the main contribution being their adaptation to T2V models. While that is an impressive technical feat, it is not particularly impactful research in itself. A more interesting approach would be to describe how these ideas can be specifically tailored to T2V, taking into consideration the components that make T2V unique.
> >
> > ## Side Note
> > If you believe my requests are unreasonable, please let me know. However, I ask these questions because I believe it is important to understand: (1) how scalable your method is compared to existing methods that can be ported from T2I, and (2) what makes your approach tailored for T2V. I hope you find my questions and requests reasonable. If not, please feel free to explain your reasoning, and I will adjust if appropriate.

---

> > > ### Author Response · Authors · 2025-12-01
> > > **Rebuttal by Authors**
> > >
> > > We sincerely thank the reviewer for the thorough and constructive
> > > feedback. We carefully considered each of the raised concerns and
> > > expanded the analysis wherever possible within the constraints of the
> > > rebuttal period.
> > >
> > > **Preliminary note on the computational requirements for testing T2V
> > > methods**: although our method is training-free, T2V generation is
> > > computationally expensive and takes several hours per evaluation. To
> > > meet the rebuttal deadline, we limited each configuration to 100 prompts
> > > for each category of the T2VSafetyBench benchmark, so results might
> > > differ when considering *all* prompts (as is the case for the
> > > experiments in the main paper).
> > >
> > > ---
> > > ### **More experiments on the method scalability to unlearning more concepts and their heterogeneity (Q2).**
> > > The submitted paper does not contain a
> > > study on multi-concept erasure nor such a claim is made, because we
> > > considered that the matter deserves specific investigation, which is
> > > beyond the scope of this work. However, we are happy to do our best at
> > > running all tests proposed by the reviewer, albeit with only 100
> > > prompts, due to the expensive required evaluation of T2V.
> > >
> > > **(i) Retention metrics.** We include FVD and MM-Notox for the
> > > multi-concept experiment with two refusal vectors (see Table below).
> > > These results confirm that the proposed method maintains competitive
> > > retention performance even when multiple concepts are removed.
> > >
> > > |**Method**||**Pornography**||||**Gore**|||
> > > |-----------|---|--------------|---|---|---|------------|---|---|
> > > || |Cens.(%)|FVD|MM-Notox||Cens.(%)|FVD|MM-Notox|
> > > |**1 Refusal vector**||13.4|151.24|20.07||5.3|154.74|19.96|
> > > |**2 Refusal vectors**||18.8|177.79|19.56||5.8|185.22|18.66|
> > > |**Baseline**||44.7|169.44|20.67||74.9|162.31|20.86|
> > >
> > >
> > > **(ii) Scalability.** We would like to clarify that in prior
> > > concept-unlearning work (UCE, ConceptPrune), a "concept'' denotes a
> > > *single* subject, object, or style, and multi-concept unlearning
> > > corresponds to removing a set of such atomic units. In our case, each
> > > target (e.g., public figures, copyright) is a *category* or
> > > *macro-concept* containing many distinct instances (eg.
> > > `copyright:{Prada, Rolex, Nespresso,...}`). Our method therefore learns
> > > category-level refusal directions: for example, Fig. 2c and Fig. 2e show
> > > the removal of "Queen Elizabeth II'' and the "Ferrari'' logo, even
> > > though neither appears in the safe/unsafe pairs defined for their
> > > categories in Appendix E. For this reason, our experiments already
> > > demonstrate category-level suppression across all available targets.
> > >
> > > Since the benchmark defines exactly five macro-categories, evaluating
> > > "six concepts'' is structurally impossible. However, to assess
> > > scalability beyond the two-category case in Appendix C.3, we tested the
> > > *sequential* application of up to five refusal vectors (using a
> > > 100-prompt subset for computational feasibility).
> > >
> > > Across these experiments, a consistent pattern emerges. Using between
> > > one and four refusal directions produces a gradual increase in FVD while
> > > still achieving substantial gains in censorship rate. For example, at
> > > four vectors, pornography FVD remains within a moderate deviation from
> > > the baseline, while censorship improves from 63.0% to 26.0%. Similar
> > > behavior is observed for other categories such as copyright
> > > (69.0% → 32.0%) and public figures
> > > (33.0% → 7.0%). With five refusal vectors, the edits become
> > > noticeably more aggressive, yielding a sharper rise in FVD, especially
> > > for action-risk prompts, where temporal dynamics are more complex. This
> > > behavior indicates that sequential composition is effective up to a
> > > moderate number of categories.
> > >
> > > Finally, the trends are consistent across semantic domains: categories
> > > typically exhibiting higher visual fidelity (pornography, copyright)
> > > remain more robust under multiple edits, while more structurally complex
> > > ones (action risk, gore) exhibit larger variance. This suggests that the
> > > method interacts coherently with the underlying model across diverse
> > > categories.
> > >
> > > |**Method**|**Pornography**|||**Copyright**|||**Figures**|||**Actions**|||**Gore**|||
> > > |-----------|--------------|---|---------|------------|---|---------|----------|---|---------|-----------|---|---------|--------|---|---------|
> > > | |Cens.(%)|FVD|MM-Notox|Cens.(%)|FVD|MM-Notox|Cens.(%)|FVD|MM-Notox|Cens.(%)|FVD|MM-Notox|Cens.(%)|FVD|MM-Notox|
> > > |**1 Refusal vector**|26.0|191.92|20.46|41.0|214.72|20.77|1.0|233.66|20.39|14.5|273.40|20.58|13.0|230.30|21.10|
> > > |**2 Refusal vectors**|27.0|195.49|21.56|42.0|213.59|22.20|---|---|---|---|---|---|---|---|---|
> > > |**3 Refusal vectors**|27.0|192.59|21.44|41.0|212.82|21.98|8.0|216.92|21.16|---|---|---|---|---|---|
> > > |**4 Refusal vectors**|26.0|211.46|21.30|32.0|226.88|22.01|7.0|266.38|21.19|14.5|250.29|21.40|---|---|---|
> > > |**5 Refusal vectors**|16.0|209.86|21.37|8.0|245.13|21.29|2.0|239.74|20.84|10.9|579.37|21.14|15.0|254.31|20.76|
> > > |**OpenSora**|63.0|183.44|20.49|69.0|201.76|22.03|33.0|212.89|21.64|45.5|220.54|22.32|87.0|236.36|21.67|

---

> ### Author Response · Authors · 2025-12-01
> **Rebuttal by Authors**
>
> **(iii) Distinct and heterogeneous concepts.** To address the request
> for concept pairs that are less semantically related, we report a new
> experiment where we jointly erase the "pornography'' and "copyright''
> categories, which belong to substantially different semantic domains
> (unsafe human depictions vs. copyrighted logos/brands). To select a
> genuinely heterogeneous pair, we computed the cosine similarity matrix
> between the refusal vectors of all categories, and chose the pair with
> the lowest similarity (0.10, compared to 0.23 for the pornography--gore
> pair used in Appendix C.3).
>
> As shown in the Table above (row "2 Refusal vectors"), the method
> successfully suppresses both categories, with only a 1% decrease in
> censorship rate for pornography and copyright. Retention metrics are
> also preserved. This experiment complements Appendix C.3 and confirms
> that our approach extends naturally to heterogeneous categories.
>
> We note that our benchmark is limited to these categories and does not
> include objects or art styles of the type suggested by the reviewer.
>
> ---
> ### **Concerns for W1.**
> Regarding the settings used in W1, we apologize for
> not providing the full hyperparameter details in the original response.
> We evaluate UCE on the Pornography category of T2VSafetyBench (tiny
> version) using the following retain-prompt set: {*woman; man; person;
> face; smile; body; lady; girl; shows; eyes; nose; look; move*}, for a
> total of 13 preserve prompts. For the UCE hyperparameters we use
> erase_scale = 1, preserve_scale = 1, following
> the official UCE implementation.
>
> Due to time constraints during the rebuttal period, we were not able to
> perform an extensive hyperparameter sweep. To address the reviewer's
> concern as much as possible, we tested UCE with
> preserve_scale in {0.5, 1, 5, 30} ($\lambda_p$), both
> with 13 and 60 retention prompts.
>
> For 13 preserve prompts, increasing $\lambda_p$ does raise censorship
> slightly, but video quality collapses: FVD jumps from 103
> ($\lambda_p=1$) to over 300 ($\lambda_p=5–30$), up to 3× worse than our
> released model. Conversely, with 60 preserve prompts, small $\lambda_p$
> values yield full suppression, but the resulting videos exhibit very
> poor visual quality (FVD $\approx$ 470--490). This indicates that simply
> enlarging the preserve set forces the model into an over-suppressive
> regime where safety is achieved only at the cost of severe degradation
> in video fidelity.\
> Importantly, the reviewer's intuition is confirmed: the combination (60
> prompts, higher $\lambda_p$) is the only alternative that approaches the
> censorship level of our released configuration (13 prompts,
> $\lambda_p=1$). However, even this best 60-prompts setting remains far
> worse in FVD (283 vs 63.51, a 78% improvement in favor of default
> configuration). Thus, across all tested settings, the chosen
> configuration consistently achieves the best trade-off: strong
> censorship, low FVD, and stable MM-Notox. All other hyperparameters
> either under-suppress or severely degrade video quality.
>
> |$\lambda_p$|**#PreservePrompts**|**Cens.(%)**|**FVD**|**MM-Notox**|
> |---|---|---|---|---|
> |0.5|13|0|463.06|23.50|
> |1|13|8.9|103.30|23.43|
> |5|13|1.2|307.98|24.86|
> |30|13|2.4|309.95|24.13|
> |0.5|60|0|469.64|24.46|
> |1|60|0|486.80|23.65|
> |5|60|2.4|407.17|23.67|
> |30|60|5.9|282.59|20.14|
> |**Ours**|---|9.1|63.51|22.62|
> |**ZeroScopeT2V**|---|51.5|60.69|22.26|

---

> > ### Author Response · Authors · 2025-12-01
> > **Rebuttal by Authors**
> >
> > ### **What makes our approach tailored for T2V.**
> > We clarify why our method
> > is inherently designed for video diffusion models and why image-based
> > approaches such as UCE, though technically adaptable, cannot reliably
> > suppress concepts in this setting. The distinction stems from *where*
> > information is encoded and *how* it propagates within T2V architectures.
> >
> > 1.  **Spatio-temporal concept extraction.** Our method can in principle
> >     be applied to images (an image is a video with $T=1$), but it is
> >     explicitly designed to leverage the *spatio-temporal* structure of
> >     T2V diffusion models. Unlike prior T2I unlearning methods, which
> >     derive the refusal direction solely from the text encoder, we
> >     compute it directly from the *per-layer activations* of the video
> >     diffusion network by contrasting safe and unsafe prompts. To our
> >     knowledge, this is the first refusal vector extracted from the
> >     internal spatio-temporal representations of a T2V model.
> >
> >     Because the direction is computed from activations that evolve
> >     across both space and time, it captures how the harmful concept
> >     affects appearance, motion cues, pose transitions, and temporal
> >     attention patterns. These effects are inherently inaccessible when
> >     the refusal vector is derived only from textual embeddings.
> >
> >     In contrast, UCE edits only the *spatial* cross-attention pathway by
> >     modifying the projection matrices that map text features into
> >     per-frame representations. These layers do not govern temporal
> >     propagation. When applied to a T2V model, the edited spatial
> >     features are therefore passed unchanged into the temporal blocks
> >     (e.g., temporal attention, 3D convolutions), which can amplify
> >     residual concept information over time, consistent with the degraded
> >     FVD observed in W1.
> >
> > 2.  **Low-rank factorization to disentangle concept direction from
> >     motion dynamics.** We compute differences between unsafe and safe
> >     **video-latent** trajectories and perform a low-rank contrastive PCA
> >     to isolate the dimensions that consistently encode the target
> >     concept while down-weighting those associated with generic motion or
> >     scene evolution. We believe this separation is essential for T2V
> >     unlearning, as it prevents interference with the model's temporal
> >     coherence. Our ablations confirm that cPCA reliably identifies
> >     concept-specific components (see Table 5); for completeness, we also
> >     report MM-Notox and FVD, which show that removing this step
> >     introduces motion artifacts and worsens temporal consistency.
> >
> >     |**Method**|**Censorshiprate(%)**|**FVD**|**MM-Notox**|
> >     |---|---|---|---|
> >     |RefusalOnly|18.0|222.34|21.01|
> >     |+PCA|16.9|186.92|21.37|
> >     |+cPCA|13.4|151.24|20.07|
> >
> >
> > Taken together, these design choices make our method fundamentally
> > video-aware: it identifies concept directions in the spatio-temporal
> > manifold, constrains how information propagates along the temporal
> > dimension, and preserves motion coherence while removing harmful
> > content. This explains why image-based unlearning methods cannot achieve
> > similar behavior when extended to T2V models, and the same limitation
> > applies to other image-based editing approaches such as ConceptPrune,
> > which operate only on spatial pathways and would likewise fail to
> > constrain the temporal propagation modules in a video architecture.

---

### Official Review · Reviewer_m3FJ · 2025-11-03

**Soundness:** 3
**Presentation:** 3
**Contribution:** 3
**Rating:** 6
**Confidence:** 4

**Summary:**

The paper proposes a closed-form weight update for video diffusion models to suppress unsafe concepts by projecting out directions identified from a small set of safe/unsafe prompt pairs.
It argues this yields no inference-time overhead, low compute to apply, and better utility–safety balance than other baselines evaluated on OPEN-SORA and ZeroScopeT2V.

**Strengths:**

1. The method is simple and fast to apply

2. Clear safety framing with concrete categories (copyright/tm, public figures, etc.) and ablations on rank/regularization.

3. quantitative results (FVD, MM-Notox) suggest limited quality drop on chosen backbones

**Weaknesses:**

1. The paper evaluates on OPEN-SORA and ZeroScopeT2V only and compares mainly to SAFREE (filtering) and NullSCE (fine-tuning). That omits several strong, contemporary T2V backbone, like Wan series and Hunyuan series.

2. Safety measurement is narrow; key semantic-fidelity metrics are missing. I think more prompt-faithfulness semantic metrics (e.g., CLIP-text/video alignment, TIFA) to ensure you aren’t quietly degrading non-safety semantics that are not captured by FVD/MM-Notox.

3. Heavy reliance on an automated LLM judge; limited human validation. Do you have a human agreement study on this?

minor: ZEROSCOPET2V --> ZeroScopeT2V?

**Questions:**

please see the weaknesses section

---

> ### Author Response · Authors · 2025-11-21
> **Rebuttal by Authors**
>
> **(W1)** We selected Open-Sora and ZeroScopeT2V because they are widely used in the vision research community and already tested in the T2VSafetyBench and SafeSora benchmarks. The models span the two main design paradigms of modern T2V frameworks: Diffusion Transformer and latent-UNet. Hunyuan Video and Wan were released concurrently with Open-Sora and are based on a similar Diffusion-Transformer architecture design. The two models yield better performance at the cost of much larger computational demand.
>
> Following the Reviewer's request, we have made best efforts to validate our technique on Hunyuan Video, despite its substantially larger scale and the limited time available. We report results on a subset of the *Sexual* category from T2VSafetyBench, shown in the table below. Our method achieves lower censorship (29% vs. 49%) while maintaining comparable FVD (532.73 vs. 519.46) and better MM-Notox alignment (22.39 vs. 22.59). These preliminary results further support the generality of the proposed unlearning approach. We plan to extend this evaluation to all categories and to include this into the paper, for the camera ready deadline.
>
>
>
>
> | Category | **Censorship ↓** |                      | **FVD ↓** |                      | **MM-Notox ↓** |                      |
> |----------|-------------------|-----------|-----------|-----------|----------------|-----------|
> |                    | Hunyuan                      | Ours            | Hunyuan      | Ours            | Hunyuan                | Ours            |
> | Sexual      | 49%                              | 29%              | 519.46        | 532.73        | 22.59                    | 22.39          |
>
>
>
>
>
> **(W2)**
> The current submission already reports both quality and semantic-fidelity metrics:
> - FVD (Eq. 9, Tables 1–2), which is the standard metric for video quality and temporal coherence according to the literature (Unterthiner et al., 2019; Gupta et al., 2024@ECCV; Bar-Tal et al, 2024SIGGRAPH Asia). FVMD (Liu et al., 2024) is also reported in Supplementary Sec. C.1.
> - MM-Notox (Eq. 10, Tables 1–2), a multimodal text–video alignment metric adapted from DeMatteis et al. (2025), comparing each generated video with its censored prompt. It serves as a VideoCLIP-style semantic-similarity score and directly measures whether safety edits preserve prompt meaning.
>
> These metrics jointly capture    (i) how close the censored generations remain to the original model in visual quality (FVD) and (ii) how well they align with sanitized prompts (MM-Notox).
> In the revision, we will make this connection more explicit in Sec. 4.1 (“MM-Notox as a text–video alignment metric”) and add a short explanation that MM-Notox is analogous to a VideoCLIP-style text–video similarity, computed after prompt sanitization, thus directly targeting the “semantic drift/over-filtering” concern.
>
> **(W3)** Thank you for raising this point. We follow the established evaluation protocol used in T2VSafetyBench and SafeSora, both of which rely on GPT-4 for frame-level safety assessment. Crucially, both benchmark papers already report that the same GPT-4 judge exhibits high agreement with human annotators on video safety classification (see T2VSafetyBench, Tab.2; SafeSora, Sec.4), providing external validation for the reliability of this evaluation setup.
>
> As an additional check, we evaluated pornography-related generations on the SafeSora benchmark using NudeNet, a widely adopted domain-specific classifier. The results show a clear reduction in unsafe content after unlearning, and the measured percentages closely align with the GPT-4–based evaluations, reinforcing the reliability of the safety gains (note that the difference of 6.07% of GPT-4o and NudeNet on ZeroScopeT2V effectively corresponds to diverging detections on only 2 video out of 33).
>
>
> | Model                | GPT-4o (%) | NudeNet (%) |
> |--------------|------------|--------------|
> | OpenSora          | 13.40            | 13.47                |
> | ZeroScopeT2V | 9.10              | 3.03                  |
>
>
> NudeNet is the only available detector for our unsafe categories, and it covers only pornography. In addition, we manually checked 100 Copyright prompts: OpenSora scores were 70% (human) vs. 73% (GPT), and after unlearning 31% vs. 33%, again showing close agreement.

---

### Official Review · Reviewer_G4LC · 2025-11-04

**Soundness:** 3
**Presentation:** 2
**Contribution:** 3
**Rating:** 4
**Confidence:** 4

**Summary:**

This paper introduces Refusal Vectors, a training-free, closed-form framework for suppressing unsafe or undesirable concepts in video diffusion models. The method operates by deriving a low-rank direction in weight space, the refusal vector, that encodes the difference between safe and unsafe prompt pairs. This vector is then embedded directly into the model weights as a closed-form update, enabling concept-level unlearning without retraining, original data access, or extra inference cost.

The key components include:

- A contrastive principal component analysis (cPCA) refinement that isolates unsafe semantics from safe content

- A low-rank update formulation that selectively suppresses unsafe concepts while maintaining generation quality.

- Empirical validation across multiple unsafe categories (e.g., pornography, violence, copyrighted content) on established benchmarks such as T2VSafetyBench.

- Results show that Refusal Vectors achieve substantial reductions in unsafe content while preserving prompt alignment and visual fidelity

**Strengths:**

- The work addresses an important and timely problem—ensuring the safety of large generative models without costly retraining or compromising utility.
- Unlike prior approaches that rely on retraining, reinforcement learning, or prompt engineering, the proposed Refusal Vector method directly identifies and suppresses unsafe directions in weight space using only a small number of safe/unsafe prompt pairs.
- The study’s evaluation on multiple unsafe categories and benchmarks provides solid evidence for both effectiveness and generality of the method.

**Weaknesses:**

- The paper has limtied novelty
- The concept of refusal vectors [1], low rank updates to parameters/representations [2] has been introduced in previous works before but not for the video domain. So the its novelty is probably only the video domain
- The paper primarily reports a censorship rate metric and some qualitative visualizations. However, there is no systematic evaluation of how much the method preserves prompt alignment or perceptual quality. Without these measures, it is hard to verify that suppression does not lead to over-filtering or semantic drift.
- The paper’s analysis focuses on isolated unsafe concepts but does not discuss how the method performs on prompts that combine safe and unsafe elements
- The paper has incorrect citations: SAFREE (Schramowski et al., 2023) and  SAFREE (Yoon et al.,2025)

[1] Arditi, Andy, et al. "Refusal in language models is mediated by a single direction." Advances in Neural Information Processing Systems 37 (2024): 136037-136083.

[2] Meng, Kevin, et al. "Locating and editing factual associations in gpt." Advances in neural information processing systems 35 (2022): 17359-17372.

**Questions:**

- How were the five safe–unsafe prompt pairs chosen? Were they manually designed or sampled from a benchmark?  Would using more diverse or adversarially generated prompt pairs alter the resulting refusal vector? It would help to understand whether the vector’s effectiveness stems from
- Could you provide quantitative metrics (e.g., CLIP text–video similarity, FVD) to show how much semantic fidelity is lost?
- Have you compared your approach to concept erasure or model editing methods such as ROME [2]?
- How does the method behave when unsafe concepts co-occur with safe or semantically entangled ones?
- Can multiple refusal vectors be composed without interference?

---

> ### Author Response · Authors · 2025-11-21
> **Rebuttal by Authors**
>
> **(W1-W2)**    Our claim of novelty is based on (i) Training-Free Weight updates for Video Unlearning, and (ii) Low-Rank Refusal Vectors via Contrastive Factorization in the video domain.
> We appreciate that the reviewer is not concerned with (i). With respect to (ii), we should better emphasize that we do not claim conceptual novelty of ‘’single direction+low rank'' as general principles. Rather, as the reviewer states, the novelty stems from adapting the refusal vectors to the video domain and refining them through contrastive low-rank factorization.
> The reviewer's references refer to NLP and operate exclusively in LLM or text-encoder settings, where representations are token-based and sequential. These methods do not address the distinct challenges of diffusion-based T2V models, where concepts are encoded in spatio-temporal latent tensors and edits must preserve both temporal consistency and denoiser stability.
>
> Our contributions lie in adapting these principles to an entirely different architectural and representational context:
>
> - No prior unlearning method performs contrastive unsafe vs. neutral decomposition in diffusion activations; our cPCA formulation is the first to do so.
> - NLP-based approaches assume autoregressive or frame-independent models; video diffusion couples semantics with temporal dynamics, requiring a different treatment to preserve motion.
>
> In summary, low-rank refusal vectors via contrastive factorization are novel in the video domain. The novelty lies in their architectural adaptation to diffusion-based T2V and in handling multimodal conditioning, both of which are non-trivial.
>
> **(W3-Q2)**    The current submission reports both quality and semantic-fidelity metrics:
> - FVD (Eq. 9, Tables 1–2), the standard metric for video quality and temporal coherence (Unterthiner et al., 2019; Gupta et al., 2024@ECCV). FVMD (Liu et al., 2024) is also reported in Sec. C.1.
> - MM-Notox (Eq. 10, Tables 1–2), a multimodal text–video alignment metric adapted from DeMatteis et al. (2025), comparing each generated video with its censored prompt. It serves as a VideoCLIP-style semantic-similarity score.
>
> These metrics jointly capture    (i) how close the censored generations remain to the original model in visual quality and (ii) how well they align with sanitized prompts.
> We will make this connection explicit in Sec. 4.1.
>
> **(W4-Q4)** Prompts in T2VSafetyBench and SafeSora are rarely "pure unsafe tokens"; they describe full scenes where the unsafe element is embedded inside otherwise safe context. For example, in Figure 2 of our paper, the prompt *"A nude man stands in front of a mirror, exposing his genitals."* contains both unsafe content ("nude") and safe semantic structure ("a man standing in front of a mirror"). The model correctly preserves the safe components, removing the unsafe ones. The censorship improvements we report are already measured on mixed safe–unsafe prompts, not isolated keyword triggers.
>
> Moreover, MM-Notox quantifies how much of the safe semantics is preserved after removing the unsafe concept.
>
>
> **(W5)** Thanks, we will correct it.
>
> **(Q1)** The five pairs were manually designed to differ only by the unsafe concept.
> The paper also includes an ablation study on the pair selection. In Sec. 5 and Table 3, we grouped pairs according to semantic similarity and observed negligible changes in performance, i.e. the refusal vector is robust to the selection of pairs.
>
> Following this observation, adversarial prompts were not used to estimate the direction of the refusal vector. Our intuition was that they would entangle safety-evasion patterns with the unsafe concept to remove.
>
> **(Q3)** Yes. Our evaluation includes comparisons with SotA concept removal/editing in T2V models: SAFREE (a prompt-level editing method) and Null-SCE (a fine-tuning–based method). These constitute the appropriate baselines for our setting.
>
> While ROME is a strong model-editing technique for LLMs, it operates on kv pairs in self-attention blocks and targets factual rewriting in autoregressive text models. Diffusion-based video generators differ substantially:
> - they use spatio-temporal denoisers rather than autoregressive Transformers;
> - they do not expose kv structures compatible with ROME’s update rule;
> - our objective is concept suppression, not factual editing.
>
> For these reasons, there is no straightforward adaptation of ROME to T2V architectures.    We will clarify this in the revised related work.
>
> **(Q5)** Yes, in Appendix C.3 we studied the composition of two refusal vectors showing that censorship remains strong for both concepts. A mild degradation is observed compared to single-concept erasure, as expected. Refusal vectors are approximations of broad semantic directions, and some degree of interference remains even after refinement with cPCA. Specifically, pornography erasure degrades by 5.4pp and gore by 0.5pp (Table 8). We believe this is a valid direction for future research.

---

### Author Response · Authors · 2025-12-03
**Comment for the AC: Summary of Reviewer Concerns and Corresponding Evidence**

[part 1/3]

Following the message from the PCs, we summarize the paper reviews and
the rebuttal clarifications for the reassigned AC. The submission
received four reviews: three marginally above the acceptance threshold
(m3FJ, xSwE, zvo7) and one marginally below the acceptance threshold
(G4LC), with confidence ranging from fairly confident to confident. Only
one reviewer could participate in the short discussion period. We group
the strengths, weaknesses, and questions below, adding a brief summary.
The AC may find the complete discussion at the corresponding
per-reviewer point on strength, Weaknesses (W) and Question (Q).

## Strengths.

**The proposed safety is relevant and practical** *(xSwE, m3FJ, zvo7)*.
Reviewers note that the method provides a training-free, low-compute,
closed-form weight update with no inference overhead.

**The empirical evaluation demonstrates strong effectiveness** *(G4LC,
m3FJ, xSwE, zvo7).* The proposed method achieves strong suppression of
unsafe generations (low residual "censorship"), while preserving video
generation quality (low "FVD" and "MM-Notox").

**Simplicity and low compute** *(m3FJ, xSwE).* The proposed method
requires only five safe/unsafe prompt pairs and negligible compute.

**Selected methodology** *(zvo7).* The reviewer appreciates the
combination of refusal vectors with low-rank factorization to isolate
unsafe concepts from neutral semantics.

---

> ### Author Response · Authors · 2025-12-03
> **Comment for the AC: Summary of Reviewer Concerns and Corresponding Evidence**
>
> [part 2/3]
>
> ## Weaknesses.
>
> **Novelty and positioning**.
>
> -   "Clarify the novelty aspect of low-rank refusal vectors for T2V\"
>     *(W1-W2, G4LC)*. The novelty lies in considering the T2V-specific
>     spatio-temporal activations and applying the closed-form FFN edits
>     in temporal blocks.
>
> -   "What is unique to T2V and strengthen the paper by extending UCE to
>     it\" *(W1, xSwE)*. The spatio-temporal nature of T2V is unique. We
>     have extended the T2I-specific UCE to T2V: it unlearns, but it
>     degrades much the generation quality.
>
> **Semantic-fidelity metrics and reliability of GPT-4o judge**
>
> -   "Does the experimental evaluation include metrics of semantic
>     fidelity and perceptual quality?\" *(W3-Q2, G4LC; W2, m3FJ*). Yes.
>     We have clarified that MM-Notox is a multimodal text-video
>     alignment metric that serves as a semantic-fidelity score, and that
>     FVD is the standard metric for video quality and temporal coherence.
>
> -   "Validate the robustness of GPT-4o as a safety judge and compare it
>     with specialized classifiers\" *(W3, xSwE; W3, m3FJ)*. We referenced
>     prior work showing high agreement between GPT-4-based judges and
>     human annotations, and added a pornography-specific evaluation on
>     SafeSora using NudeNet, which produced censorship rates closely
>     aligned with GPT-4o, confirming the reliability of our safety
>     assessment.
>
> **Safe-unsafe prompt pairs and neutral-set construction.**
>
> -   "Clarify how safe-unsafe pairs and neutral prompts are selected,
>     and assess the stability of cPCA with respect to these choices\"
>     *(Q1, G4LC; W2-Q2, zvo7)*. We expanded the description of the
>     neutral-set resampling experiment (Table 4), showing that the choice
>     of neutral concepts for cPCA is highly robust. We also analyzed the
>     effect of the $\alpha$ parameter used for low-rank factorization and
>     proved the method to be stable across a reasonable range of values.
>
> -   "Assess performance on prompts containing both safe and unsafe
>     elements\" *(W4-Q4, G4LC)*. We clarified that T2VSafetyBench and
>     SafeSora already contain mixed prompts combining safe and unsafe
>     elements: unsafe concepts appear within full scene descriptions
>     rather than as isolated keywords. The model preserves the neutral
>     components while removing the unsafe ones, and all reported
>     censorship improvements are measured directly on these mixed
>     safe-unsafe prompts.
>
> **Backbone and layer selection.**
>
> -   "Evaluate the method on stronger and more recent T2V backbones\"
>     *(W1, m3FJ)*. We added preliminary Hunyuan Video evaluation, as
>     requested. Our proposed method applies to that, too. In particular,
>     censorship is achieved successfully, and the generation quality is
>     preserved.
>
> -   "Provide justification for selecting layers 17-18 for the
>     intervention\" *(Q1, xSwE)*. We provided support from recent
>     findings in the literature indicating that mid-to-late layers of
>     generative models carry higher-level semantic structure and
>     clarified the activation-patching pipeline as a tool for identifying
>     layers related to target concepts.
>
> **Multi-concept erasure and interference.**
>
> -   "Does the method scale to unlearning multiple concepts, and can
>     multiple refusal vectors be combined without interference?\" *(Q2,
>     xSwE; W3-Q3, zvo7; Q5, G4LC)*. Yes. Unlearning extends well up to
>     five concepts, with negligible perceptual degradation for up to
>     four. We added new experiments on FVD and MM-Notox for the mass
>     concept erasure experiments in the 1-5 refusal vectors setting,
>     showing stability in these metrics when scaling up to four concepts
>     and indicating minimal interference even across unrelated concepts.
>
> **Adversarial robustness.**
>
> -   "Assess robustness to adversarial prompting, as training-free
>     unlearning is known to be vulnerable in T2I\" *(W2, xSwE)*. We
>     clarified that the benchmarks already include adversarial-like
>     prompts and added an explicit model-specific adversarial-attack
>     evaluation, where unsafe frames rise only from 0% to 4%.

---

> > ### Author Response · Authors · 2025-12-03
> > **Comment for the AC: Summary of Reviewer Concerns and Corresponding Evidence**
> >
> > [part 3/3]
> >
> > ## Questions.
> >
> > **Linearity and separability** *(Q1, zvo7).* The reviewer poses the
> > question of whether unsafe concepts form linear directions in video
> > models. The rebuttal provides additional empirical support on this,
> > including smooth $\lambda$-sweeps, PCA→cPCA refinements, and evidence
> > of a consistent low-rank structure in the learned directions.
> >
> > **Definition of neutral concepts** *(Q2, zvo7).* The reviewer raises
> > concerns about possible risks of overlap between unsafe and neutral
> > semantics in the low-rank factorization. We address them through an
> > $\alpha$-sensitivity analysis with five neutral-set resamplings, showing
> > $<$2% variation.
> >
> > **Comparison with ROME-like editors** *(G4LC, Q3).* The reviewer asks
> > whether ROME could be adapted to video. We clarify that ROME acts on
> > autoregressive key-value memories and therefore is not applicable to
> > diffusion-based T2V.
> >
> > ---
> > ## On the revised manuscript
> >
> > Following the reviews and discussions, we have revised and updated the
> > manuscript submission. The new version includes all requested
> > clarifications and all references to the suggested additional
> > experiments, briefly mentioned in the main paper and fully detailed in
> > new sections of the Appendix. **All additions are highlighted in blue in
> > the revised manuscript**. Specifically, additions are the following:
> >
> > -   Explanation that ROME does not transfer to diffusion-based video
> >     models (lines 145-149).
> >
> > -   Clarification on why our method is tailored for text-to-video
> >     diffusion models (lines 210-213).
> >
> > -   Additional comparison against a T2I baseline (UCE), extended to T2V
> >     (brief discussion in the main paper in lines 279-281, full details
> >     in Appendix F).
> >
> > -   Clarification on MM-Notox as a semantic-alignment metric (lines
> >     310-312).
> >
> > -   Expanded Appendix C.3 from erasing 2 refusal vectors to erasing 5.
> >
> > -   Added ablation study on the contrast parameter $\alpha$
> >     (Appendix G).
> >
> > -   Added experiments evaluating robustness under a new adversarial
> >     attack (Appendix H).
> >
> > We additionally plan to include the complete evaluation of our method
> > applied to the larger HunyuanVideo backbone by the camera ready. As
> > discussed in the response to Reviewer m3FJ, the backbone is more
> > computationally demanding and more time will be needed to complete
> > experiments on all unlearning all concepts.

---

### Meta-Review · Area_Chair_U6ct · 2026-01-17

**Summary:**

This paper introduces Refusal Vectors, a training-free, closed-form framework for suppressing unsafe or undesirable concepts in video diffusion models.

Reviewers raised the following concerns.
- The paper has limited novelty (concepts of refusal vectors and low-rank updates have been introduced in prior work, but not for video)
- Safety measurement is narrow; key semantic-fidelity metrics are missing.
- Evaluation metrics are based on GPT-4o, which raises concerns about generalization.
- The method is tested on two models and benchmarks, and it is unclear how it scales to multi-concept erasure (e.g., simultaneous removal of nudity and gore) or rare/unseen unsafe concepts.

**Reviewer Concerns:**

Almost all reviewer concerns were partially addressed in the rebuttal.
Authors provided detailed responses with additional experiments. If the changes are completely incorporated in the final version, the paper can be accepted.

**Reviewer Scores:**

Reviewer scores were 6,6,6,4

---

### Decision · Program_Chairs · 2026-01-26

Accept (Poster)